# Understanding of Cervical Screening Adoption among Female University Students Based on the Precaution Adoption Process Model and Health-Belief Model

**DOI:** 10.3390/ijerph20010700

**Published:** 2022-12-30

**Authors:** Hye Young Shin, Purum Kang, Soo Yeon Song, Jae Kwan Jun

**Affiliations:** 1Department of Nursing, Gangseo University, Seoul 07661, Republic of Korea; 2National Cancer Control Institute, National Cancer Center, Goyang 10408, Republic of Korea; 3College of Nursing, Woosuk University, Wanju 55338, Republic of Korea; 4Graduate School of Cancer Science and Policy, National Cancer Center, Goyang 10408, Republic of Korea

**Keywords:** Pap smear, uterine cervical neoplasms, young adult, health-belief model, university student, decision-making

## Abstract

This study aimed to identify the decisional stages for cervical screening and corresponding cognitive factors in female university students. A cross-sectional study was conducted among Korean female university students aged 20–29 years through an online survey. The main outcome was the decisional stage of cervical screening adoption using the precaution adoption process model. The stages were classified into pre-adoption (the unawareness, unengaged, and undecided stages), adoption (the decided to act and acting stages), and refusal (the decided not to act stage). The cognitive factors in each stage were analyzed using the key concepts of the health-belief model. Cervical screening was defined as a clinical Papanicolaou (Pap) smear performed by a physician in a clinic. The final analysis included 1024 students. Approximately 89.0%, 1.0%, and 10.2% were classified as the pre-adoption, refusal, and adoption stages, respectively. Compared to the pre-adoption stage group, the adoption stage group was more likely to be older and have higher self-efficacy and knowledge. Most participants belonged to the pre-adoption stage—those unaware of cervical screening. Furthermore, most had a low level of knowledge and perception of cervical cancer and its screening. Therefore, our study highlighted the need for education to increase awareness and knowledge of cervical screening in this population.

## 1. Introduction

Since the Papanicolaou (Pap) smear test is a relatively easy and non-invasive tool for cervical cancer screening; it is generally recommended as a primary test for cervical screening [1,2]. In Korea, approximately 26.0% of young women in their 20s have undergone cervical screening using a Pap smear, which was significantly lower than that in other age groups (56%) [3].

The precaution adoption process model (PAPM) is an important theoretical framework that explains a person’s decision-making process to avoid health threats through precautious behaviors [4]. Compared with the behavior change stage model, the PAPM is more applicable for individuals with low awareness of health behaviors since it adds decisional stages that do not recognize health behavior (Stages 1 and 2, unaware and unengaged by the issue, respectively) and the stage wherein no action was decided upon (Stage 4, decided not to act) [5,6]. Moreover, individuals’ beliefs, self-efficacy, and knowledge influence the movement of the stage towards the adoption of health behaviors [7,8,9]. In the context of cancer-screening participation, PAPM has been used to identify individuals’ decisional stages related to breast [5], colorectal [8,10], and cervical cancer screening [6,7,9,11]. Regarding cervical cancer screening, in a study of UK women who did not participate in cervical cancer screening, 28% of them were unaware of the screening; particularly, young women aged 25–34 years compared with those aged over 35 years were significantly unaware of the screening [6]. The study confirmed that there was a difference in health beliefs at each stage [7]. In a qualitative study, the barriers to participation in cervical cancer screening at each stage of PAPM were evaluated [11].

The health-belief model (HBM) is a theoretical framework that explains and predicts health-related behaviors [12]. The key concepts of the HBM are based on individual beliefs, which consist of perceived susceptibility, severity, benefits, and barriers, and self-efficacy [12]. Previous studies have identified significant differences regarding individual beliefs by HBM that affect cervical screening participation [9,13]. Among the key concepts, self-efficacy is the perception of self-control ability [14]; particularly, a direct relationship has been observed between self-efficacy and the likelihood of undergoing cervical screening [15]. In addition, knowledge is a modifying factor of the HBM and affects individuals’ health beliefs and participation in cervical screening [16]. In an Iranian study, housewife women were more likely to participate in cervical screening as they perceived more benefits and fewer barriers to screening and had higher self-efficacy for undergoing screening [17]. In addition, the determinants of interest in cervical screening participation were perceived threats, benefits, and barriers, and cues for action [18].

Currently, there are only a few studies regarding the non-participation of young women in cervical screening in Korea [19,20]. Therefore, we aimed to identify the decisional stage for cervical screening adoption and cognitive factors associated with each stage among female university students.

## 2. Materials and Methods

### 2.1. Study Design and Participants

In Korea, a descriptive and cross-sectional study was conducted between August and September 2015. All participants were recruited through online community sites of four universities located in Seoul, South Korea. This study included female students aged 20–29 years who are attending the university and have no history of cervical cancer. An online survey was conducted for data collection. The study descriptions and written orientations regarding the risks of participating in the study were posted in the online communities of the universities. Participants expressed their consent by ticking the consent box and answering the questionnaire. A total of 1250 participants answered the online questionnaire; however, 226 participants who used the same internet protocol address (IP address) were excluded. Therefore, 1024 participants were included in the final analysis. The study was evaluated by STROBE (strengthening the reporting of observational study cross-sectional study). This study was approved by the Institutional Review Board of the NCC, Korea (IRB No. NCC2015-0066).

### 2.2. Data Collection and Outcome Measures

Survey items were developed based on a review of previous studies and the theoretical frameworks of the PAPM and HBM. To ensure the content validity of the questionnaire, three specialists in obstetrics and gynecology, laboratory medicine, and preventive medicine, as well as two nursing professors, participated in modifying and confirming the survey items. In our study, cervical screening was defined as a clinical Papanicolaou (Pap) smear performed by a physician in a clinic. To identify the participants’ characteristics, relevant questions were classified into four categories: (1) perception of cervical cancer and screening based on HBM; (2) knowledge regarding cervical cancer and its prevention; (3) the decisional stage of cervical screening adoption based on PAPM; and (4) preference regarding cervical sampling (clinician-sampled Pap smear) or vaginal sampling (self-sampled human papilloma virus (HPV) test).

The perception of cervical cancer and screening based on HBM consisted of five subscales: perceived susceptibility, perceived severity, perceived benefits, perceived barriers, and self-efficacy regarding cervical cancer. Additionally, the screening was measured using the champion health-belief model scale [21]; on the other hand, self-efficacy was measured through the replacement of breast cancer with cervical cancer [22]. Among the items of the perceived barrier subscale, one item was deleted (I feel uncomfortable talking about cervical cancer) due to its inappropriateness for young women with less exposure to cancer. Furthermore, two items were added based on previous research [23]: ‘I prefer a female doctor to conduct a cervical screening’ and ‘There is no health care center close to my house to have a cervical screening’. Regarding the self-efficacy items, two items were excluded that were related to the arrangement of transportation and of knowledge of cervical screening procedures. This study was conducted in a large city; therefore, the means of transportation were abundant. The other item was related to knowledge instead of to self-efficacy. All of the items (perceived susceptibility (5 items, Cronbach’s alpha = 0.862), perceived severity (7 items, Cronbach’s alpha = 0.830), perceived benefits (6 items, Cronbach’s alpha = 0.916), perceived barriers (7 items, Cronbach’s alpha = 0.734), and self-efficacy (8 items, Cronbach’s alpha = 0.863)) were rated on a five-point Likert scale ranging from strongly disagree to strongly agree.

Knowledge about cervical cancer was measured using 11 items (four items for risk factors; two for symptoms; two for prevention; and three for epidemiology and treatment). The items on risk factors consisted of questions about the relationship between HPV infections, transmission through sexual intercourse, sexual intercourse at an early age, smoking, and ‘cervical cancer’. Regarding symptoms, items about early and late symptoms of cervical cancer were included. Considering prevention, items about preventable cervical cancer and participation in the Pap smear in sexually experienced women were included. Questions about epidemiology and treatment included the incidence rate of cervical cancer, the importance of early detection, and HPV vaccination. Each item could be scored as 1 for correct responses or 0 for incorrect or ‘I don’t know’ responses. The possible scores ranged from 0 (poor knowledge) to 11 (high knowledge). Cronbach’s alpha for the knowledge scale was 0.863.

PAPM was used to measure the decisional stage of cervical screening adoption [4]. Stage 7 (maintaining up-to-date screening status) of PAPM was not considered, due to the low age of the sample, which might decrease the need to undergo regular check-ups for cancer. The stages were classified into three groups: (1) The pre-adoption stage group, which included stages 1 (unawareness; uninformed regarding cervical screening), 2 (unengaged; aware of cervical screening but has not provided serious thoughts regarding screening), and 3 (undecided; considered but has not decided to undergo cervical screening); (2) The refusal stage group, which included stage 4 (decided not to act; decided to undergo cervical screening); and (3) The adoption stage group, which included stages 5 (decided to act; decided to undergo cervical screening) and stage 6 (acting; underwent cervical screening). To evaluate the preference between cervical (clinician-sampled Pap smear) and vaginal sampling (self-sampled HPV test), explanations regarding the two methods are presented with images of vaginal sampling.

### 2.3. Statistical Analysis

All statistical analyses were performed using SPSS version 25.0 software (IBM Corp., Armonk, NY, USA). Differences between groups were analyzed by Student’s *t*-test or χ^2^ test. Statistical significance was set at *p* < 0.05. Logistic regression analysis was conducted to identify the associations between potential factors and the decisional stages of cervical screening adoption of PAPM.

## 3. Results

### 3.1. General Characteristics and Decisional Stage of Cervical Screening Adoption

The general characteristics of the participants are shown in Table 1. A total of 1024 participants were included in the study, with a mean age of 22.65 ± 2.48 years. Among them, 89.0% were classified into the pre-adoption stage (unawareness, 68.8%; unengaged, 6.3%; and undecided, 13.9%). On the other hand, 10.2% were classified into the adoption stage, among whom 5.4% (decide to act) of the participants decided to undergo cervical screening and 4.8% (acting) had undergone cervical screening.

In the refusal stage (decided not to act), only 1.0% of participants decided not to undergo cervical screening. Table 1 shows the distribution of participants by stage based on the PAPM.

### 3.2. Factors Related to Decisional Stage of Cervical Screening Adoption

The characteristics of the study participants according to the decisional stages of cervical screening adoption based on PAPM and HBM are presented in Table 2. Few participants (n = 10) decided against undergoing cervical screening (stage 4); therefore, they were excluded from the bivariate analysis. Multiple logistic regression analysis was conducted to identify factors associated with the decisional stages of cervical screening adoption. The adoption stage group was significantly older (adjusted odds ratio (aOR): 1.13, 95% confidence interval (CI): 1.03 to 1.23), had higher self-efficacy (aOR: 1.13, 95% CI: 1.07 to 1.20), and had higher knowledge (aOR: 1.36, 95% CI: 1.23 1.51) compared with the pre-adoption stage group.

### 3.3. Preference of Cervical Screening Method

To identify the preferred screening methods for each stage group, data were analyzed using the χ^2^ test. Compared with the adoption stage group, participants in the pre-adoption stage preferred vaginal sampling when performing cervical screening (χ^2^: 7.84, *p* < 0.01) (Figure 1).

## 4. Discussion

This study identified the decisional stages of cervical screening adoption and factors associated with the stages of PAPM and HBM in female university students. Most participants were classified into the pre-adoption stage; on the other hand, approximately, only 10.0% were included in the adoption stage. Participants in the adoption stage had an increased likelihood of having higher knowledge of cervical cancer and screening and had higher self-efficacy for receiving cervical screening than those in the pre-adoption stage.

Based on the PAPM, our study confirmed that most participants were in the pre-adoption stage (stages 1-3) of cervical screening. In particular, 68.8% of the participants had no knowledge or were unaware of cervical screening (stage 1); this is consistent with the study of Kim [20] in Korean female university students. On the other hand, 4.8% of participants in our study received cervical screening, which was significantly lower than the 20.9% and 54.7% cervical screening rate of women in their 20s in Korea and Japan, respectively [3,24]; furthermore, approximately 68.0% of sexually active university students in the United States have received cervical screening [25]. The Korean Guideline for cervical cancer screening recommends a Pap smear every 3 years in women aged 20–74 years [1]. Since 2016, the National Cancer Screening Program has been providing cervical cancer screening using a Pap smear every two years in women over the age of 20 [1]. Nevertheless, as a result of analyzing trends in cervical cancer incidence and mortality in Korea over the past 20 years, it has significantly increased only in young women in their 20s [26]. Therefore, active community-based efforts are necessary for encouraging young women to participate more in cervical cancer screening.

Regarding cognitive factors for the decisional stages of cervical screening adoption based on HBM, the adoption stage group showed higher self-efficacy than the pre-adoption stage. In cancer screening, self-efficacy is an important determinant of the screening rate [15], which predicts the decisional stage progression for cervical screening [16]. In South Africa, a previous study reported that female university students who underwent cervical screening had significantly higher levels of self-efficacy than those who had not undergone cervical screening [27]. Moreover, a high level of self-efficacy for human papillomavirus (HPV) vaccination and sexual health behaviors with condom use was related to increased intention of HPV vaccination and condom usage among university students [28]. However, we found that an individual’s belief of HBM, except self-efficacy, was slightly higher in the adoption stage group than in the pre-adoption stage, but the difference was not significant. In particular, low perceived susceptibility for cervical cancer was evident in both stages, which may be attributed to misconceptions regarding cervical cancer caused by a lack of knowledge and awareness in young women (i.e., they are healthy or they are not at risk of cervical cancer) [20,24].

Additionally, knowledge is a modifying factor of HBM that influences individual beliefs and health behaviors [16]. In our findings, knowledge of cervical cancer and prevention was significantly higher in participants in the adoption stage compared than in those in the pre-adoption stage. However, most participants were classified as the pre-adoption stage; therefore, there was a greater number of participants who had an unsatisfactory level of knowledge regarding cervical cancer and prevention. This finding is consistent with previous studies in different countries. In India, a survey of female university students found that approximately 98.0% of participants were unaware of cervical screening, and approximately 95.0% were uninformed regarding the risk factors of cervical cancer [29]; studies in Malaysia and UK showed similar results [30,31].

Previous studies have shown that HBM-based educational interventions increased the screening rate [32]; furthermore, the strategies for each decisional stage of cervical screening adoption using a behavioral stage change model yielded positive results in women of all ages [9,33]. These educational interventions were similarly effective in enhancing cervical screening rate in female university students [34]. Therefore, we suggest the use of HBM-based educational interventions to improve the knowledge and beliefs of female university students regarding cervical screening. In addition, we found that participants in the pre-adoption stage preferred vaginal sampling (self-sampled HPV test) compared to cervical sampling (clinician-sampled Pap smear). Therefore, the approach of using vaginal sampling could be considered to be an effective method to engage non-participants [35].

This study has several limitations. First, this was a cross-sectional study that did not establish a clear and causal relationship between antecedent and consequent factors. Second, the sexual activity of the participants were not evaluated. Kim [36] reported that the sexual experience of female university students was approximately 60.0%, which indicates a high possibility that participants with no sexual experience were included. Therefore, in our study, the number of participants in the adoption stage of cervical screening would have been lower. Nevertheless, our study focused on the population with the lowest cervical screening rate among the screening target age groups: female university students in their early 20s. Furthermore, it was the first to identify the decision stages of the cervical screening adoption and cognitive factors according to stage. Therefore, our findings could contribute to the establishment of interventions for improving cervical screening by providing an in-depth understanding of behaviors related to screening among young women.

## 5. Conclusions

In conclusion, our study confirmed that most female university students belonged to the pre-adoption stage: those who were not aware of cervical screening and rarely underwent screening. Moreover, participants in the pre-adoption stage were less likely to have lower knowledge and self-efficacy than those in the adoption stage. Therefore, our findings highlight the need for a decisional stage-based educational approach to increase cervical screening adoption among female university students.

## Figures and Tables

**Figure 1 ijerph-20-00700-f001:**
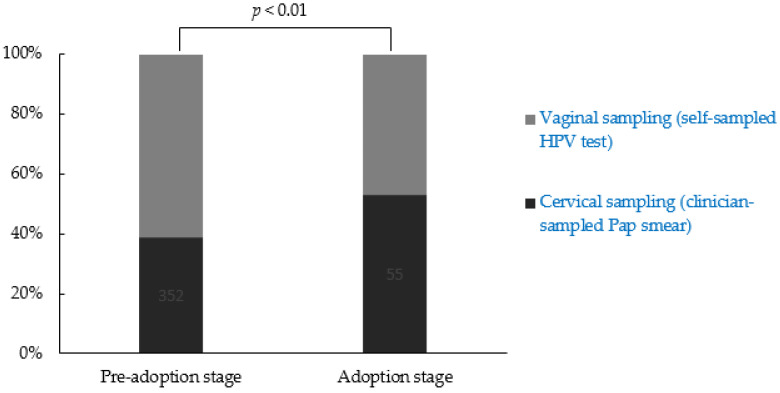
Preference in terms of screening methods according to the decisional stage of cervical screening adoption. The pre-adoption stage group, which included stages 1 (unawareness), 2 (unengaged), and 3 (undecided); and the adoption stage group, which included stages 5 (decided to act) and stage 6 (acting).

**Table 1 ijerph-20-00700-t001:** General characteristics of participants.

Characteristics		Total (n = 1024)
Age, mean (SD)		22.65 (2.48)
Health behavior model, mean (SD)		
Perceived susceptibility (range 5–25)		11.46 (3.33)
Perceived severity (range 7–35)		19.80 (4.67)
Perceived benefits (range 6–30)		23.91 (3.63)
Perceived barriers (range 7–35)		21.80 (4.23)
Self-efficacy (range 8–40)		32.60 (5.16)
Knowledge (range 0–11), mean (SD)		6.40 (2.60)
Family history for cervical cancer, n (%)		
No		960 (93.8)
Yes		64 (6.3)
Gynecological treatment in the past, n (%)		
No		856 (83.6)
Yes		168 (16.4)
Smoking, n (%)		
No		968 (94.5)
Yes or Ever		56 (5.5)
Drinking, n (%)		
No		316 (30.9)
Yes		708 (69.1)
HPV vaccination, n (%)		
No		627 (61.2)
Yes		397 (38.8)
Decisional stage based on PAPM, n (%)		
Pre-adoption stage	Unawareness (stage 1)	704 (68.8)
	Unengaged (stage 2)	64 (6.3)
	Undecided (stage 3)	142 (13.9)
Refusal stage	Decided not to act (stage 4)	10 (1.0)
Adoption stage	Decided to act (stage 5)	55 (5.4)
	Acting (stage 6)	49 (4.8)

Abbreviations: HPV, human papillomavirus; SD, standard deviation; and PAPM, precaution adoption process model.

**Table 2 ijerph-20-00700-t002:** Associated factors related to decisional stages of cervical screening adoption based on PAPM stage.

Characteristics	Pre-Adoption	Adoption	*p-* Value	Adoption vs. Pre-Adoption		Adoption vs. Pre-Adoption	
	n = 910	n = 104		cOR	(95% CI)	aOR	(95% CI)
Age, mean (SD)	22.53 (2.42)	23.60 (2.70)	<0.001	1.18	(1.09–1.28)	1.13	(1.03–1.23)
Health behavior model, mean (SD)							
Perceived susceptibility (range 5–25)	11.42 (3.29)	11.85 (3.57)	0.219	1.04	(0.98–1.10)	1.05	(0.97–1.12)
Perceived severity (range 7–35)	19.80 (4.56)	20.04 (5.40)	0.667	1.01	(0.97–1.06)	1.00	(0.95–1.06)
Perceived benefits (range 6–30)	23.87 (3.60)	24.46 (3.73)	0.121	1.05	(0.99–1.11)	0.93	(0.86–1.01)
Perceived barriers (range 7–35)	21.97 (4.15)	20.55 (4.66)	0.001	0.93	(0.88-0.97)	0.96	(0.90-1.02)
Self-efficacy (range 8-40),	32.29 (5.01)	35.61 (5.20)	<0.001	1.16	(1.11-1.21)	1.13	(1.07-1.20)
Knowledge (range 0–11), mean (SD)	6.18 (2.56)	8.25 (2.13)	<0.001	1.43	(1.30–1.57)	1.36	(1.23–1.51)
Family history for cervical cancer, n (%)							
No	860 (94.5)	92 (88.5)	0.015	1.00	(Reference)	1.00	(Reference)
Yes	50 (5.5)	12 (11.5)		2.24	(1.15–4.37)	1.60	(0.77–3.33)
Gynecological treatment in the past, n (%)							
No	770 (84.6)	79 (76.0)	0.024	1.00	(Reference)	1.00	(Reference)
Yes	140 (15.4)	25 (24.0)		1.74	(1.07–2.83)	1.47	(0.86–2.49)
Smoking, n (%)							
No	862 (94.7)	97 (93.3)	0.535	1.00	(Reference)	1.00	(Reference)
Yes or Ever	48 (5.3)	7 (6.7)		1.30	(0.57–2.94)	1.56	(0.65–3.78)
Drinking, n (%)							
No	278 (30.5)	35 (33.7)	0.516	1.00	(Reference)	1.00	(Reference)
Yes	632 (69.5)	69 (66.3)		0.87	(0.56–1.33)	0.96	(0.59–1.54)
HPV vaccination, n (%)							
No	564 (62.0)	56 (53.8)	0.107	1.00	(Reference)	1.00	(Reference)
Yes	346 (38.0)	48 (46.2)		1.40	(0.93–2.10)	0.88	(0.56–1.38)

Abbreviations: aOR, adjusted odds ratio; CI, confidence interval; cOR, crude odds ratio; HPV, human papillomavirus; SD, standard deviation; and PAPM, precaution adoption process model. The pre-adoption stage included participants who were unaware (stage 1), unengaged (stage 2), and undecided stage (stage 3); the adoption stage included the participants who decided to act (stage 5) and those who were acting (stage 6).

## Data Availability

The dataset supporting the conclusions of this article is available from the corresponding author on reasonable request.

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
