# Peer review of "Understanding of Cervical Screening Adoption among Female University Students Based on the Precaution Adoption Process Model and Health-Belief Model"

_ijerph, 2022, doi:10.3390/ijerph20010700_

Round 1

Reviewer 1 Report

Introduction

(1) The statement Currently there are only a few studies regarding the non-participation of young women in cervical screening in Korea  (page 2, line 56) should be referenced (those few studies should be referenced)

(2) Why is the data from 2015 - it is 7 years old?  

(3) Define or operationalized cervical screening.  Does it refer to clinical pap smears?   HPV testing?  Both?   The study narrative, particularly in the Introduction focuses on cervical cancer and in the methods the statement is made To evaluate the preference between doctor-based cervical sampling (Pap smear) and vaginal self-sampling, explanations regarding the two methods were presented with pictures for self-sampling.   A clinical PAP smear done by a physician the cells in the cervical swap would be examined.  The home test (self-sampling) would primarily be screening for HPV.   I could  be wrong here but I think it needs clarification (what is cervical screening?  the types? what can be done at home?   etc. to better contextualize the paper) 

Methods

(4) Please address the validity and reliability of the survey.

(5) Was the survey piloted (focused field test)?  Revised?  Panel of Experts used?

Author Response

We thank you for taking the time and effort necessary to review our manuscript and provide us with these valuable comments and suggestions. Accordingly, we revised our manuscript and made changes to it. Please, note that our changes to the manuscript are highlighted in blue font for your own convenience.

Reviewer 2 Report

The purpose of this study was to examine the cervical screening adoption among female university students based on two classical health theories. Overall, the method is clear and appropriate, and the results are solid. I would like to point out the following as an attempt to further improve the manuscript contents:

1.      The introduction need to be expended. For example, which is the relationship between Pap smear test and cervical screening? Are they the same thing? What is the difference between those two?

2.      Why is it important for female students in their 20s to undergo cervical screening? Please add more evidence to support the discussion, such as medical guidelines from health organizations or authorities.

3.      Although there are limited studies to examine the cervical cancer screening adoption, I believe there are many literatures using these two theories to investigate women’s awareness about cervical screening in other areas, such as U.S. and Europe. A more thoroughly literature review is needed in the introduction.

4.      Method: what does the knowledge items looks like? Maybe provide several examples or add a table to list all items.

5.      Any psychological test be performed to test the validity and reliability of the theory constructs? (Cronbach alpha and CFA)

6.      Discussion: another limitation of this study is that the data was collected in 2015. The situation may be changed because now is 2022, seven years past.

7.      Conclusion: line 232-233, this sentence is confusing. Please avoid using a double negative in the sentence.

Author Response

(The authors gave the same response as above.)

Round 2

Reviewer 2 Report

Thanks for addressing all questions.